# Association between Walking Habit and Physical Frailty among Community-Dwelling Older Adults

**DOI:** 10.3390/healthcare10081396

**Published:** 2022-07-27

**Authors:** Tsubasa Yokote, Harukaze Yatsugi, Tianshu Chu, Xin Liu, Hiro Kishimoto

**Affiliations:** 1Department of Behavior and Health Sciences, Graduate School of Human-Environment Studies, Kyushu University, Fukuoka 819-0395, Japan; tsubasayokote@icloud.com (T.Y.); chutianshu_japan@yahoo.co.jp (T.C.); liuxinjp1992@gmail.com (X.L.); 2Faculty of Arts and Science, Kyushu University, Fukuoka 819-0395, Japan; haru19920424@gmail.com; 3Center for Health Science and Counseling, Kyushu University, Fukuoka 819-0395, Japan

**Keywords:** physical frailty, walking, older adult, cross-sectional study

## Abstract

The aim of this cross-sectional study was to determine whether older adults who practice walking have a lower risk of physical frailty than those who do not. The study subjects were 846 older adults and were not certified as needing support or nursing care. The subjects were classified as being physically frail or pre-frail or being robust, according to the revision of the Cardiovascular Health Study criteria. We classified the subjects by questionnaire into a no-exercise group, walking-only group, walking plus other exercise group, and exercise other than walking group. In logistic regression analyses, the odds ratio (OR) and 95% confidence interval (95%CI) were shown. Compared to the no-exercise group, the OR (95%CI) for physical frailty was 0.85 (0.48–1.49) for the walking-only group, 0.54 (0.36–0.83) for the walking plus other exercise group, and 0.67 (0.47–0.97) for the exercise other than walking group. In the components of physical frailty, the walking plus other exercise group and the exercise other than group had significantly lower ORs for exhaustion. Older adults who only practiced walking as an exercise do not have lower risks of physical frailty and pre-frailty. Older adults who combine walking with other exercises or practice non-walking exercises have lower risks of them.

## 1. Introduction

Japan has the highest aging rate in the world, and the rate is expected to continue to rise. Accordingly, the costs of social security benefits in Japan, including pensions, medical care, and welfare, are reported to be on the rise [1]. Among the costs of social security benefits, the rate of increase in long-term care costs is remarkable [2]. However, there are no sufficient social environment and social security systems that enable older people to extend their healthy longevity at this time.

Frailty is one of the factors leading to the need for long-term care; it is defined as a state of decreased physiological reserve and increased vulnerability to stress in old age [3]. Frailty has various aspects, and the severity of physical frailty, among other aspects, is directly related to the need for long-term care [4]. Chen et al. reported that older adults with physical frailty at baseline had a significantly higher risk of requiring long-term care within ~6 years compared to robust individuals [5]. The prediction of physical frailty is thus important to reduce the need for long-term care and extend healthy longevity.

The amount of exercise and physical activity can be objectively expressed in terms of frequency, duration, and metabolic equivalents (METs). Japan’s Ministry of Health, Labour, and Welfare recommends that older adults should be physically active for ≥10 METs/hour per week [6]. The World Health Organization recommends ≥ 300 min per week of aerobic physical activity [7]. A review of randomized controlled trials reported that exercise or physical activity is the most effective intervention for preventing frailty [8]. However, the specific optimal exercise modalities for preventing frailty—such as the specific exercise type(s), frequency, and duration — are not clear. While considering these factors, it is necessary to establish one or more methods that can achieve the recommended amount of exercise that can be practiced continuously by many older adults and thus contribute to reducing the risk of physical frailty.

Among the many types of exercise, our research group has focused on walking. In community-dwelling older adults, it is reported that those who spend more time walking have a better life expectancy [9]; the practice of aerobic exercise such as walking improves carotid compliance [10], and older adults with a walking habit have higher physical function. Walking is safe, low-cost, and easily disseminated, and these advantages may contribute to the postponement of physical frailty by regular walking. However, to the best of our knowledge, there have been no investigations of the association between walking and physical frailty. The relatively few studies of the relationship between walking and physical function did not consider the influence of physical activity levels other than walking [11].

We conducted the present study to determine whether older adults who walk for exercise have a lower risk of physical frailty than those who do not. We also investigated exercise types, duration, and frequency in a population with a lower risk of physical frailty and pre-frailty compared to a population not engaging in exercise.

## 2. Materials and Methods

### 2.1. Study Design and Subjects

This was a cross-sectional study using baseline survey data from an epidemiological cohort study of community-dwelling older adults in Itoshima City, Fukuoka, Japan. The participants were men and women aged 65–75 years who lived in this city and responded to a community needs survey conducted in September 2016 [12] and who were not certified as requiring nursing support or care. Considering the size of the district, 5000 respondents were randomly selected and mailed the survey invitation and questionnaire. We included the 949 respondents who participated in the measurement sessions from September to December 2017 [13]. A final total of 846 subjects (399 men and 447 women) who met the criteria for valid data on physical activity (≥4 days and ≥10 h of wear time) of a tri-axis accelerometer (Active style Pro HJA-350IT, Omron Healthcare, Kyoto, Japan) and had no missing data on physical frailty or exercise habit were included in the final analysis of this study (Figure 1). This study was approved by the Institutional Review Board of Kyushu University (Ref No. 201708), and informed consent was obtained from all subjects involved in the study.

### 2.2. Measures

#### 2.2.1. Physical Frailty

Physical frailty was defined according to the revision of the Cardiovascular Health Study (CHS) criteria, using the definition applied in the Sasaguri Genkimon study [3,14]. The CHS criteria consist of five items: shrinking, exhaustion, weakness, slowness, and low physical activity. Table 1 shows a specific assessment method for the five items. Weakness was assessed using BMI and maximum grip strength. The grip strength was measured using a Smedley grip strength meter (GRIP-D, T.K.K. 5401; Takei Scientific Instruments Co. Ltd., Niigata, Japan) in an upright posture with the arm directly down, left and right, and the maximum value was used. Slowness was evaluated based on height and maximum walking speed results at 5 m. To measure walking speed, an 11 m walking path was created, and the subjects were instructed to walk a distance of 11 m at maximum speed from a stationary standing position. The time was measured from 3rd to 8th meter. Exhaustion and shrinking were measured with a questionnaire. Low physical activity was assessed by measuring physical activity energy expenditure (kcal) using a triaxial accelerometer. The subjects were instructed to wear the accelerometer on the right or left side of their waist for seven consecutive days and remove it only before going to bed or participating in water activities.

#### 2.2.2. Walking Assessment

We divided the subjects into four groups: the no-exercise group (N group), the walking-only group (W group), the walking plus other exercise group (W + O group), and the exercise other than walking group (O group). For the determination of each subject’s exercise habit, we asked “Have you exercised in the past month?”. Another question asked the subject to select up to 3 of the 19 exercise activities commonly performed by elderly Japanese, and we investigated their exercise frequency per week and the duration per session for each session of exercise. The 19 activities include Japanese sampo (i.e., strolling, including dog walking), walking, hiking, cycling, gymnastics (including stretching), yoga, golf, hitting practice golf balls, ground golf, table tennis, ballroom dancing, aerobic dancing, tai chi, underwater exercise, volleyball, catch-ball, bowling, muscle strength training, and others.

#### 2.2.3. Other Assessment

We obtained the following information from each subject via questionnaire: age, sex, body mass index (BMI), medical history (presence/absence of osteoporosis, hypertension, dyslipidemia, diabetes, cerebral infarction, cardiac disease, and other diseases, respectively), number of pain locations (presence/absence of each shoulder, elbow, wrist, hip, knee, ankle, waist, and neck), years of education, activities of daily living (ADLs; eating, moving, toileting, walking, climbing stairs, dressing, defecating, urinating, bathing, and grooming), residence area (selected from the city’s residential districts: Maebaru, Nijo, and Shima), smoking habit (selected from “almost every day”, “sometimes”, “used to smoke but quit”, and “never”), and alcohol consumption (selected from “almost every day”, “sometimes”, “rarely”, and “never”). The subjects’ cognitive function was measured with the Mini-Mental State Examination (MMSE). If a subject was engaging in more than one exercise type, we used the average duration per exercise type (minutes per session). The frequency of exercise per week (number of times per week) was calculated by adding up the number of exercise days per week for each exercise if the subject performed more than one exercise type. Sedentary time (ST; activity intensity < 1.5 METs), light physical activity (LPA; activity intensity of 1.5 METs to <3.0 METs), and moderate-to-vigorous physical activity (MVPA; activity intensity of ≥3.0 METs) were measured with the above-mentioned tri-axis accelerometer.

#### 2.2.4. Statistical Analyses

The subjects’ characteristics are expressed as the mean ± standard deviation (SD) or *n* (%), and a one-way analysis of variance (ANOVA) or χ^2^ test was performed. A post hoc Bonferroni test was used for multiple comparisons among the four exercise groups. Physical frailty and physical pre-frailty were set together as “at risk”, and a binary variable between that status and the robust status was created and used for logistic regression analysis. The analysis was used to calculate the odds ratio (OR) and 95% confidence interval (CI) for comparing the N group as the reference group with the other three groups. The risk of each component of physical frailty was also examined by a logistic regression analysis. Adjustment factors in the logistic regression analysis were those reported to be associated with physical frailty in previous studies: age, gender, BMI, number of medical histories, number of pain locations, years of education, a total ADL score, the MMSE score, smoking habit (almost daily or occasional smoking), drinking habit (almost daily or occasional drinking), and the three residential districts. Moreover, MVPA and ST were also included as adjustment factors to exclude the effects of daily living activities other than walking and of sitting time during exercise breaks and non-exercise time. The distribution of exercise types in the W + O and the O groups was determined. The distribution of frequency of exercise per week and duration of exercise per session were also assessed, including the W group, and comparisons between groups were tested by the post hoc Bonferroni test. The statistical analyses were conducted using Statistical Analysis Software (SAS) ver. 9.4 (SAS Institute, Cary, NC, USA). The computation was carried out using the computer resource offered under the category of General Projects by the Research Institute for Information Technology, Kyushu University. The statistical significance level was set at *p* < 0.05.

## 3. Results

### 3.1. The Caracteristics of Exercise Behavior Groups

A summary of the four exercise groups’ characteristics is provided in Table 2. Compared to the no-exercise/reference N group, the W group had significantly higher MMSE scores and more MVPA. The W + O group had significantly fewer pain locations, longer years of education, and more MVPA. The W + O group exercised significantly more frequently than the W and O groups. The mean exercise duration was significantly longer in the O group than in the W and W + O groups.

### 3.2. Association between Exericise Behavior and Physical Frailty or Pre-Frailty

The results concerning the relationship between exercise and physical frailty or pre-frailty and its components are given in Table 3. In the N group, 114 of the 212 subjects were physically frail or pre-frail. In the W group, 33 of the 75 subjects (44.0%) were physical frailly or pre-frail. The OR (95%CI, *p*-value) for physical frailty or pre-frailty in the W group compared to the N group was 0.77 (0.44–1.33, *p* = 0.35), with no significant difference. On the other hand, in the W + O group, 71 of the 204 subjects (34.8%) were physically frail or pre-frail, with a significantly lower OR at 0.51 (0.33–0.77, *p* = 0.0014). Similar results were observed in the O group (OR: 0.66, 95%CI: 0.46–0.94, *p* = 0.02), in which 146 of the 355 subjects (41.2%) were physically frail or pre-frail. Regarding the association with the components of physical frailty, the W + O group had a significantly lower OR at 0.30 (95%CI: 0.15–0.59, *p* = 0.0005) for exhaustion and 0.20 (95%CI: 0.06–0.71, *p* = 0.01) for slowness. The O group also had a significantly lower OR for exhaustion (OR: 0.51, 95%CI: 0.31–0.84, *p* = 0.008) and a lower trend for slowness (OR: 0.47, 95%CI: 0.22–1.01, *p* = 0.054).

Table 4 shows the results of the association between exercise and physical frailty or pre-frailty after the addition of ST and MVPA as adjusting factors. The W group showed no significant association with physical frailty or pre-frailty (OR: 0.85, 95%CI: 0.48–1.49, *p* = 0.57). This relationship was unchanged from before the adjustment for these two factors. The risk of physical frailty or pre-frailty in the W + O group also showed no change in association and was significantly lower than that in the N group (OR: 0.54, 95%CI: 0.36–0.83, *p* = 0.0048). The risk in the O group was also significantly lower than that in the N group (OR: 0.67, 95%CI: 0.47–0.97, *p* = 0.03). Regarding the association with the components of physical frailty, the significantly lower OR of exhaustion that we observed in the W + O and O groups did not change after adjusting for ST and MVPA (OR: 0.30, 95%CI: 0.15–0.59, *p* = 0.0006 for the W + O group, and OR: 0.51, 95%CI: 0.31–0.85, *p* = 0.009 for the O group). In contrast, the OR for slowness in both W + O and O groups was attenuated; it was marginal in the W + O group and no longer significantly associated in the O group.

### 3.3. Distribution of Exercise Types Other Than Walking

The most common types of exercise used together in the W + O group were gymnastics (including stretching) at 46.1% and strength training at 15.2%. The most common exercise activities for the O group were sampo (including dog walking) at 37.7% and gymnastics (including stretching) at 37.5%.

## 4. Discussion

The results of our analyses demonstrated that the risk of physical frailty or pre-frailty in the W(alking) group was not significantly different from that in the N(o-exercise) group. On the other hand, the risk of physical frailty or pre-frailty was significantly lower in the W + O and O(ther) groups compared to the N group. This relationship remained unchanged after adjustment for the subjects’ sedentary time and moderate-to-vigorous physical activity. The lower risk found in the W + O and O groups was due to a significantly lower risk of exhaustion, a component of physical frailty.

These results contradict our hypothesis that the subjects with a walking habit alone would not be at a lower risk of physical frailty. The reasons for this outcome could be due to walking’s low intensity of exercise, the little amount of exercise duration per session and the low frequency of exercise per week in the W group, and the lack of statistical power due to the small sample size of the group (*n* = 75). It has been reported that older adults with a walking habit had significantly longer 6 min walk test distances [15]. The American College of Sports Medicine guidelines indicate that exercise tolerance improves by 10% with exercise intensity equivalent to ≥60% of exercise tolerance [16]. However, it has been reported that the walking intensity of community-living older adults is 40% of their maximum oxygen uptake [17]. It is therefore possible that walking alone may not reach the intensity necessary to improve exercise tolerance, which is associated with exhaustion, a component of physical frailty.

Regarding the frequency of exercise, the older adults who engaged in regular walking only in this study had a lower frequency of exercise per week than the other groups. It has been reported that walking improves depression, anxiety, psychological health, and mental health [18] and that more frequent exercise was associated with lower depressive symptoms [19]. In intervention studies, healthy individuals who exercised more frequently had significantly higher rates of improvement in muscle strength and muscle mass [20,21]. Since we used the Kessler-6, a method of assessing psychological stress [22], to assess exhaustion, we speculate that the risk of exhaustion was not lowered by the lack of positive effects of infrequent exercise on mental and muscles.

The duration of exercise per session was shorter in the W group than in the other groups, which is also characteristic of older adults who engage in walking exclusively. In a 10-year prospective follow-up study, older women who walked ≥40 min/day had a lower risk of developing depression than older women who walked <30 min/week [23]. A cross-sectional study of older adults showed a lower risk of depression with greater total exercise time per week [24], and a cross-sectional study of subjects aged 19–89 years showed that the longer the exercise duration, the higher the maximal oxygen uptake [25]. It is thus possible that the duration of exercise per session in our W-group subjects was insufficient to improve mental and exercise tolerance.

In this study, the risk of exhaustion was lower among the subjects who combined walking with other exercises. In a cross-sectional study, the subjects who combined aerobic physical activity with strength training had significantly lower depressive symptoms compared to engaging in either of these activities alone [26]. Gymnastics and strength training were the most common disciplines used in combination by the present study’s subjects. Earlier investigations revealed that participating in gymnastics classes improved quality of life, *ikigai* (positive attitude) [27], and exercise self-efficacy [28], suggesting that the combined use of gymnastics has a positive impact on mental health. Lower-limb muscle strength, a factor associated with exercise tolerance, is improved by strength training and gymnastics [29,30]. The presence of these in combination with walking in our W + O subjects may thus have resulted in better mental status and muscle strength plus reduced exhaustion.

Engaging in multiple types of exercise also significantly increased the frequency of exercise compared to the W and O groups. We suspect that the greater frequency of exercise may have contributed to the lower risk of exhaustion. This association was also observed when the analysis was adjusted for ST and MVPA, suggesting that the combination of walking and other exercises in itself contributes to the reduction in risk of exhaustion regardless of the sedentary time or the degree of moderate-to-vigorous intensity exercise or lifestyle activity. The risk of exhaustion was also lower among the present older adults who engaged in exercise other than walking. Many of the subjects in this group were walking or doing gymnastics. As noted above, participating in gymnastics has a positive effect on mental health [27,28] and reduces exhaustion. Although *sampo* is similar to walking, the subjects who used the more leisurely *sampo* pace had a longer duration of exercise per session, which may have contributed to their lower risk of exhaustion.

To the best of our knowledge, this is the first study to focus on walking and its association with physical frailty. The older adult subjects were analyzed separately as those who engage in only a walking habit and those who used walking plus other physical activities. The analysis excluded the influence of a number of factors, including the subjects’ objectively measured ST and MVPA. Exercise activities other than walking and the exercises’ frequency and duration were evaluated to identify the characteristics of the exercise group that was at a low risk of developing physical frailty. However, there are several study limitations to address: (*1*) This was a cross-sectional study, and the causal relationship between walking and physical frailty is unclear. (*2*) The small number of people in the W group (*n* = 75) may have resulted in insufficient statistical power. (*3*) The intensity of exercise, including walking, was unknown in this study. (*4*) Depending on the exercise category, the amount of time that the subjects reported engaging in exercise may have included rest time. However, the results were adjusted for sedentary time, and thus, the subjects’ break time during exercise is not likely to have affected the results. In this study, exercise habit was not associated with the components of physical frailty: grip strength, weight loss, and decreased activity. Therefore, these components require further investigation, such as combining them with factors other than exercise. It is also necessary to clearly assess the intensity of exercise itself in community-dwelling older adults.

## 5. Conclusions

Older adults who only practiced walking as an exercise do not have lower risks of physical frailty and pre-frailty. However, our findings suggest that engaging in a high frequency of walking in combination with other exercises or maintaining a longer duration of exercise other than walking is associated with a lower risk of exhaustion and a lower risk of physical pre-frailty or frailty.

## Figures and Tables

**Figure 1 healthcare-10-01396-f001:**
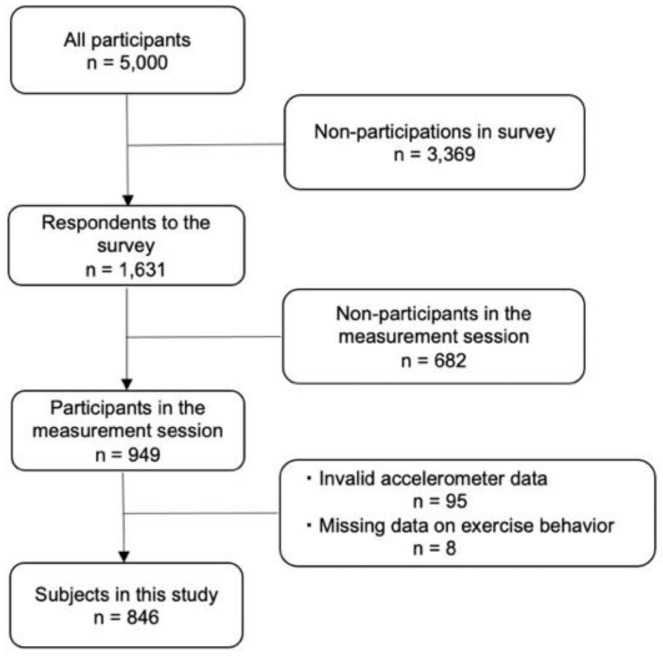
Participant flow chart in the analysis.

**Table 1 healthcare-10-01396-t001:** Definition of physical frailty.

Shrinking	Unintentional weight loss > 2–3 kg in the prior 6 months
Weakness	Grip strength stratified by gender and BMI (kg/m^2^)
Men	≤25.00 kg for BMI < 18.5, ≤30.00 kg for 18.5 ≤ BMI < 25, ≤31.50 kg for 25 ≤ BMI < 30, ≤33.00 kg for BMI ≥ 30
Women	≤17.50 kg for BMI < 18.5, ≤19.50 kg for 18.5 ≤ BMI < 25, ≤20.50 kg for BMI 25 ≤ BMI < 30, ≤19.75 kg for BMI ≥ 30
Exhaustion	Positive answer to either of two self-reported questions. Participants were asked how they felt during the past 30 days: “Did you feel that everything you did was an effort?” and “Did you feel exhausted without any reason?”
Slowness	Time of 5 m walk test at one’s maximum waking speed stratified by gender and standing height (gender-specific cutoff: a medium height).
Men	Time ≥ 3.56 s for height < 162.0 cm or Time ≥ 3.21 s for height ≥ 162.0 cm
Women	Time ≥ 4.25 s for height < 148.7 cm or Time ≥ 3.61 s for height ≥ 148.7 cm
Low physical activity	Energy expenditure of physical activity by a triaxial accelerometer; quantified as kilocalories/kg (body weight), stratified by gender.
Men	≤6.20 kcal/kg/day
Women	≤7.13 kcal/kg/day
Overall frailty status	Robust: 0 affected component. Pre-frailty: 1–2 affected components. Frailty ≥ 3 affected components.

**Table 2 healthcare-10-01396-t002:** The characteristics of the four exercise behavior groups of community-dwelling older Japanese adults (*n* = 846).

	N Group *n* = 212 (25.1%)	W Group *n* = 75 (8.9%)	W + O Group *n* = 204 (24.1%)	O Group *n* = 355 (42.0%)
Age, y	70.8 ± 3.1	70.9 ± 3.3	71.1 ± 3.2	70.9 ± 3.0
Men	95 (44.8)	34 (45.3)	101 (49.5)	169 (20.0)
Residential district:				
Shima	55 (25.9)	12 (16.0)	41 (20.1)	86 (24.2)
Nijo	57 (26.9)	23 (30.7)	51 (25.0)	91 (25.6)
Maebaru	100 (47.2)	40 (53.3)	112 (54.9)	178 (50.1)
BMI, kg/m^2^	23.0 ± 3.3	23.0 ± 3.2	22.7 ± 2.9	22.9 ± 3.2
Presence of disease	99 (46.7)	34 (45.3)	104 (51.0)	159 (44.8)
No. of pain sites	1.9 ± 2.1	1.3 ± 1.6	1.4 ± 1.7 *	1.7 ± 1.8
Education, y	12.5 ± 2.3	12.7 ± 2.2	13.3 ± 2.6 *	13 ± 2.3
ADL score	10.1 ± 0.6	10.1 ± 0.6	10.1 ± 0.5	10.1 ± 0.3
MMSE score	27.5 ± 2.2	28.3 ± 1.7 *	28.0 ± 2.1	27.8 ± 2.1
Sedentary time (minutes/day)	440.6 ± 121.2	454.5 ± 111.3	445.2 ± 102.4	445.0 ± 103.9
Moderate and Vigorous physical activity, min/day	44.6 ± 33.6	67.4 ± 33.5 *	59.4 ± 30.0 *	50.6 ± 31.0
Light physical activity, min/day	350.1 ± 96.8	339.9 ± 110.3	336.1 ± 88.8	343.3 ± 90.1
Alcohol consumption, any level	101 (47.6)	33 (44.0)	113 (55.4)	180 (50.7)
Smoking habit, any level	23 (10.9)	4 (5.3)	9 (4.4)	25 (7.0)
Frequency of exercise, ×/week	0	4.8 ± 2.4	7.9 ± 4.3 ^†,§^	5.0 ± 3.6
Duration of exercise, min/session	0	73.7 ± 57.4	91.2 ± 54.5	106.3 ± 75.9 ^†,‡^

Data are the number of subjects (percentage) or mean ± standard deviation. N group, no-exercise group; W group, walking only group; W + O group, walking plus other exercise group; O group, exercise other than walking group. * *p* < 0.05 vs. N group, ^†^ *p* < 0.05 vs. W group. ^‡^ *p* < 0.05 vs. W + O group. ^§^ *p* < 0.05 vs. O group. Presence of disease, the percentage of those who have at least one of the following: osteoporosis, hypertension, dyslipidemia, diabetes, stroke, heart disease, or other disease; Alcohol consumption, the percentage of respondents who answered “I drink almost every day” or “I drink sometimes”; Smoking habit, the percentage of respondents who answered “I smoke almost every day” or “I smoke sometimes”; ADL, activities of daily living; BMI, body mass index; MMSE, Mini-Mental State Examination.

**Table 3 healthcare-10-01396-t003:** The relationship between exercise behavior and physical frailty or pre-frailty and its components.

	N Group	W Group	W + O Group	O Group
Physical frailty or pre-frailty, *n* (%):	114 (53.8)	33 (44.0)	71 (34.8)	146 (41.1)
OR	1.00	0.77	0.51	0.66
95%CI	Ref.	0.44–1.33	0.33–0.77	0.46–0.94
*p*-value	–	0.35	0.0014	0.02
Components of physical frailty:				
Shrinking, *n* (%):	31 (14.6)	7 (9.3)	23 (11.3)	35 (9.9)
OR	1.00	0.63	0.84	0.67
95%CI	Ref.	0.26–1.51	0.46–1.53	0.40–1.14
*p*-value	–	0.29	0.57	0.14
Exhaustion, *n* (%):	45 (21.2)	8 (10.7)	13 (6.4)	40 (11.3)
OR	1.00	0.51	0.30	0.51
95%CI	Ref.	0.22–1.19	0.15–0.59	0.31–0.84
*p*-value	–	0.12	0.0005	0.008
Weakness, *n* (%):	40 (18.9)	14 (18.7)	31 (15.2)	62 (17.5)
OR	1.00	1.04	0.84	0.95
95%CI	Ref.	0.52–2.10	0.49–1.46	0.59–1.51
*p*-value	–	0.91	0.54	0.82
Slowness, *n* (%):	18 (8.6)	2 (2.7)	3 (1.5)	15 (4.3)
OR	1.00	0.37	0.2	0.47
95%CI	Ref.	0.08–1.70	0.06–0.71	0.22–1.01
*p*-value	–	0.20	0.01	0.054
Low physical activity, *n* (%):	29 (13.6)	6 (8.0)	16 (7.8)	41 (11.6)
OR	1.00	0.66	0.54	0.96
95%CI	Ref.	0.24–1.78	0.26–1.12	0.55–1.69
*p*-value	–	0.41	0.10	0.89

N group, no-exercise group; W group, walking only group; W + O group, walking plus other exercise group; O group, exercise other than walking group. Adjusted for age, sex, BMI, number of diseases, number of pain locations, years of education, the total ADL score, the total Mini-Mental State Examination score, residential district, smoking habit, and alcohol consumption. OR, odds ratio; CI, confidence interval.

**Table 4 healthcare-10-01396-t004:** The relationship between exercise and physical frailty or pre-frailty and its components after adjusting for sedentary time (ST) and moderate-to-vigorous physical activity (MVPA).

	N Group	W Group	W + O Group	O Group
Physical frailty or pre-frailty, *n* (%):	114 (53.8)	33 (44.0)	71 (34.8)	146 (41.1)
OR	1.00	0.85	0.54	0.67
95%CI	Ref.	0.48–1.49	0.36–0.83	0.47–0.97
*p*-value	–	0.57	0.005	0.03
Components of physical frailty:				
Shrinking, *n* (%):	31 (14.6)	7 (9.3)	23 (11.3)	35 (9.9)
OR	1.00	0.58	0.81	0.66
95%CI	Ref.	0.24–1.44	0.44–1.48	0.39–1.13
*p*-value	–	0.24	0.48	0.13
Exhaustion, *n* (%):	45 (21.2)	8 (10.7)	13 (6.4)	40 (11.3)
OR	1.00	0.52	0.30	0.51
95%CI	Ref.	0.22–1.23	0.15–0.59	0.31–0.85
*p*-value	–	0.14	0.0006	0.009
Weakness, *n* (%):	40 (18.9)	14 (18.7)	31 (15.2)	62 (17.5)
OR	1.00	1.25	0.92	0.99
95%CI	Ref.	0.60–2.58	0.52–1.61	0.62–1.59
*p*-value	–	0.55	0.76	0.97
Slowness, *n* (%):	18 (8.6)	2 (2.7)	3 (1.5)	15 (4.3)
OR	1.00	0.55	0.28	0.54
95%CI	Ref.	0.12–2.61	0.07–1.02	0.25–1.18
*p*-value	–	0.45	0.053	0.12
Low physical activity, *n* (%):	29 (13.6)	6 (8.0)	16 (7.8)	41 (11.6)
OR	1.00	2.91	2.22	1.83
95%CI	Ref.	0.86–9.90	0.86–5.69	0.89–3.75
*p*-value	–	0.09	0.10	0.10

N group, no exercise group; W group, walking only group; W + O group, walking plus other exercise group; O group, exercise other than walking group. Adjusted for age, sex, BMI, number of diseases, number of pain locations, years of education, the total ADL score, the total Mini-Mental State Examination score, residential district, smoking habit, alcohol consumption, sedentary time, and moderate-to-high-intensity physical activity.

## Data Availability

Data sharing not applicable.

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
