# Peer review of "Association between Walking Habit and Physical Frailty among Community-Dwelling Older Adults"

_healthcare, 2022, doi:10.3390/healthcare10081396_

Round 1

Reviewer 1 Report

This study is a valuable finding investigating the effects of walking and other exercises on physical frailty. However, there are some improvements that need to be made.

1. I don't think that "There was no association between walking and physical frailty."  on line 23th and "Walking was not associated with physical frailty." on line 298 are not accurate expressions. The reason is that the risk was reduced when both walking and other exercises were performed, and there was no significant difference only when walking only. Please consider it.

2. The expression using "CHS criteria (for example, Line 15-16th)" may require reconsideration. The assessment of physical frailty this time is based on the new standard developed by the authors' research group with reference to the original CHS criteria. It should be properly stated that it is a proprietary revision of the CHS criteria, as it is likely to mislead the reader. Also, I think it will help the reader's understanding if you describe the reasons and merits of using this criterion.

3. Finally, accelerometer data is used to determine physical frailty, but in this study the MVPA evaluated by the accelerometer is adjusted at the same time. In addition, MVPA seems to be related to walking and other exercises targeted in this study. It may seem like an over-adjustment, and I think it would be helpful for the reader to explain why MVPA and ST should be adjusted.

 4. I think it's a careless mistake, but the sentences from lines 228 to 236 and lines 237 to 245 are exactly the same. Please correct.

Reviewer 2 Report

It was not associated with physical frailty, as the stated objective. However, the findings suggest the importance of promoting physical exercise for a lower risk of exhaustion and a lower risk of pre-frailty or physical frailty in our elders, with what I suggest of great interest, for approaching and improving the quality of life of elderly people.

 It is probable that the analysis was either entirely the most appropriate, or some study variable was missing, as you mentioned.

The composition of the ethics committee for the approval of the protocol is not described.

it is recommended and necessary assess the intensity of exercise itself in older adults living in the community.

I recommend explaining the data collection procedure, and better explaining the instruments used

Round 2

Reviewer 1 Report

I argue that the revised manuscript has reached a level where it can be published.